# Osteoporosis in Men: A Review of an Underestimated Bone Condition

**DOI:** 10.3390/ijms22042105

**Published:** 2021-02-20

**Authors:** Giuseppe Rinonapoli, Carmelinda Ruggiero, Luigi Meccariello, Michele Bisaccia, Paolo Ceccarini, Auro Caraffa

**Affiliations:** 1Orthopaedic and Traumatology Department, University of Perugia, Ospedale S.Maria della Misericordia, S. Andrea delle Fratte, 06156 Perugia, Italy; michelebisa@yahoo.it (M.B.); paoloceccarini84@gmail.com (P.C.); auro.caraffa@unipg.it (A.C.); 2Orthogeriatric Service, Geriatric Unit, Institute of Gerontology and Geriatrics, Department of Medicine, University of Perugia, Ospedale S.Maria della Misericordia, S. Andrea delle Fratte, 06156 Perugia, Italy; carmelinda.ruggiero@unipg.it; 3Department of Orthopaedics and Traumatology, AORN San Pio “Gaetano Rummo Hospital”, via R.Delcogliano, 82100 Benevento (BN), Italy; drlordmec@gmail.com

**Keywords:** bone fragility, fractures, male, prevention, screening, DXA, BMD

## Abstract

Osteoporosis is called the ‘silent disease’ because, although it does not give significant symptoms when it is not complicated, can cause fragility fractures, with serious consequences and death. Furthermore, the consequences of osteoporosis have been calculated to weigh heavily on the costs of health systems in all the countries. Osteoporosis is considered a female disease. Actually, the hormonal changes that occur after menopause certainly determine a significant increase in osteoporosis and the risk of fractures in women. However, while there is no doubt that women are more exposed to osteoporosis and fragility fractures, the literature clearly indicates that physicians tend to underestimate the osteoporosis in men. The review of the literature done by the authors shows that osteoporosis and fragility fractures have a high incidence also in men; and, furthermore, the risk of fatal complications in hip fractured men is higher than that for women. The authors report the evidence of the literature on male osteoporosis, dwelling on epidemiology, causes of osteoporosis in men, diagnosis, and treatment. The analysis of the literature shows that male osteoporosis is underscreened, underdiagnosed, and undertreated, both in primary and secondary prevention of fragility fractures.

## 1. Introduction

According to the WHO criteria, osteoporosis is defined as a BMD that lies 2.5 standard deviations or more below the average value for young healthy women (a T-score of <−2.5 SD) [1].

Osteoporosis is a disease widely spread all over the world, which causes a large number of fragility fractures. It is usually a silent disease unless fragility fractures occur (in that case it is classified as ‘severe’), with serious consequences. Bone fragility due to osteoporosis can cause a wide range of fractures, including, first of all, hip and vertebral fractures. However, also the fractures of the proximal humerus, wrist, ribs, ileus and ischiopubic branches, those of the ankle and other can be consequences of bone fragility related to osteoporosis. Worldwide, osteoporosis causes more than 9 million fractures a year, meaning there is a fragility fracture every 3 s [2]. In a 2013 study [3], the authors reported the cost of fragility fractures in the 27 countries of the EU. It has been estimated at €37 billion in 2010, with 26,300 life years lost.

A wide review by Ballane et al. conducted in 2014 [4] ascertained the increasing rates of hip fractures in much of Asia, Southern Europe, and South America, and predicted that there would be a greater increase in hip fractures in developing countries. A recent review published by Thambiah and Yeap [5] confirmed the increasing trend in fracture rate in developing countries. The authors’ conclusions are that BMD in Southeast Asia (SEA) populations is generally lower than in Caucasians, but hip fracture rates are not higher, although projected to increase, while vertebral fracture rates in SEA are similar to those in North America.

The burden of osteoporosis is enormous. Fragility fractures of all types can lead to serious consequences and even death. Not infrequently, they cause a significant impairment in the quality of life, decreased mobility, and increased risk of long-term care admission and mortality. Among fragility fractures with the highest incidence, hip fractures lead to a mortality of 15–20%, while vertebral fractures lead to sequelae such as chronic pain, balance disorders, digestive, and respiratory disorders [6,7].

It is an indisputable fact that osteoporosis mainly affects women: women aged 50 years or older have a four times higher rate of osteoporosis and a two times higher rate of osteopenia compared with men [8]. It is estimated that one out of every three women and one out of every five men will experience fragility fractures at least once in their lifetime [9]. The NHANES 2005–2008 study, that evaluated the bone density of the hip and lumbar spine, showed that the prevalence of osteopenia and osteoporosis at either site was 38% and 4% for men compared with 61% and 16% for women, respectively [10]. This data must not lead to underestimate osteoporosis in men, which is instead a serious problem and carries great risks. Even in the literature, there are few studies that refer specifically to osteoporosis in men.

## 2. Osteoporosis in Men—Epidemiology

What emerges from the literature is that male osteoporosis is underestimated, underdiagnosed, and undertreated.

The NHANES 2005–2006 study [11] was conducted by the National Center for Health Statistics (NCHS) on 3157 American adults aged 50 years and older, by analyzing BMD data at the femur neck and total hip. The results revealed that 49% of older women and 30% of older men had osteopenia at the femur neck, whereas 10% of women and 2% of men had osteoporosis at this site. In another study carried out in South Korea from 2008 and 2010 [12], the authors reported the BMD of 2305 male subjects aged 50–79, submitted to DXA (dual-energy X-ray absorptiometry) of total femur, femoral neck, and lumbar spine. Proportions of osteoporosis at the total femur, femoral neck, and lumbar spine were 0.7, 3.3, and 7.0%, respectively. In another study in men aged 69 to 74 [13], it emerged that osteoporosis was present in 10.2% of the sample. In fact, the prevalence of osteoporosis is difficult to calculate with precision, because of the low number of subjects who undergo DXA for screening. We will discuss this topic in a following section.

### 2.1. Fragility Fractures in Men

In 2008, as many as 109,000 men suffered osteoporotic fractures in the US [14]. Data from 2005 indicate that, of the 2 million osteoporotic fractures that occur annually in the US, 29% are men. This percentage corresponds to the associated US $ 17 billion in costs [15]. In men, the incidence of hip fractures range from 0.56 per 1000 per year at the age of 60 to 13 per 1000 per year in subjects aged 85 was found [14]. These data were extracted from the U.S. 2008 Nationwide Emergency Department Sample. Similar results were reported by Diamantopoulos et al., in a Norwegian study covering the years 2004 and 2005: an incidence of 0.49 per 1000 person years of hip fractures at age 60, 12.3 hip fractures per 1000 person years in subjects aged 85 was found [16]. It has been calculated that around 80,000 men break their hips every year [14]. In a 22-year follow-up study conducted at the Skåne University Hospital, Malmö, Sweden, on 226 hip fractured men, one in three died within a year after the injury and another one-third underwent a new fracture [17].

The risk of a hip fracture in elderly men is 5–6%, compared to 16–18% for women. This would mean that 30% of all hip fractures affect men [18,19,20]. This represents a higher percentage than commonly believed.

In 2025, hip fractures are expected to increase by 89% worldwide compared to 2000, with about 800,000 fractures per year [21]. The risk of at least one fragility fracture in a 50-year-old man was calculated to be 13% (versus 40% for women), and 25% in an 80-year-old man [22,23,24,25]. A prospective observational study showed that the relative risk (RR) of low-impact trauma fractures is higher in men than in women aged 60 years or older [25]. For what concerns the age of highest incidence of fractures, in a prospective 12-year study carried out in the Dubbo study (Australia), although men have a higher risk of hip fractures after the age of 80, half of hip fractures in men occur before that age [26]. An epidemiological observation showed how, in women, the incidence of vertebral fractures increases rapidly after the age of 55, while this phenomenon occurs in men after the age of 65. As for hip fractures, the incidence shows a rapid growth of its curve after 65 in women and after 75 in men [27].

As regards the risk of falling, sustain more fall related injuries women tend to fall more than men and sustain more injuries related to falls than men [28]. An epidemiological study by Sattin et al. [29] carried out in Miami, Florida, revealed that, in subjects aged 85, the risk of falling is 138.5 per 1000 for men and 156.8 per 1000 for women. According to the data reported by Stevens et al. [30], obtained from the National Electronic Injury Surveillance System-All Injury Program (NEISS-AIP), the annual rates of non-fatal injuries due to falls for women were 48.4% higher than the rates for men (in 2005, average 5466.7 non-fatal falls per 100,000 population for women, average 3674 non-fatal falls per 100,000 population for men).

### 2.2. Mortality

Scientific evidence shows that men have more osteoporosis-related complications and a higher mortality rate after osteoporosis fractures than women starting at the time of the hospital admission due to hip fractures [31,32,33,34]. Several authors confirm that, in the first year after a hip fracture, there is a higher mortality in men than in women [35,36,37,38], but the greatest mortality in men is also present after a vertebral fragility fracture [29]. In the prospective cohort from the Dubbo Osteoporosis Epidemiology Study of community-dwelling women and men aged 60 years and older from Dubbo, Australia [25], the authors reported that, in women, there were 952 low-trauma fractures followed by 461 deaths (48.4%), and in men, 343 fractures were followed by 197 deaths (57.4%) within 10 years. Kim reported a 6% lifetime risk of hip fracture in men [14]. The epidemiological data by Bow et al., showed that, in the Asian race, the lifetime risk is about half compared to the Caucasian race [39]. In fact, the causes for which mortality in men is higher than in women have not been clearly demonstrated. Infection is one possible explanation for the observed mortality rate [40].

Other studies [41,42] about mortality rate in osteoporotic men are summarized in Table 1.

## 3. Differences Between Male and Female Bone

The reason why women have a higher risk of fragility fractures than men is that women have bones with a smaller diameter, an earlier bone resorption process and a higher risk of fall [43].

By examining the differences between male and female bone, we can understand the differences in the bone development during the skeletal maturation and the process occurring in the skeletal aging.

### 3.1. Skeletal Maturation

Seeman [44] points out that DXA cannot be used to evaluate the geometric characteristics of bone. DXA allows us to evaluate the areal BMD and not the volumetric BMD. If the areal BMD differs according to age and sex, the volumetric remains constant. Therefore, if the bone in males is stronger, this depends exclusively on the larger size of the bone in the male, not on the greater density. Bone is stronger in men because men have a larger skeleton. However, this is not the cause of the fact that men fracture less. In fact, the strength of the bone is adequate for weight and height.

During growth, there is a prominent formation of periosteal bone in boys, especially due to the action of androgens, growth hormone (GH) and insulin-like growth factor (IGF-1). In girls, conversely, the formation of endosteal bone prevails, as estrogens inhibit the formation of periosteal bone [45,46].

### 3.2. Skeletal Aging

In men, trabecular bone loss begins already in the young adulthood, while cortical bone loss begins later, usually after the age of 50 [47]. The physiological bone loss in men and women shows a different pattern. Trabeculae in men tend to become thinner than in women, but they keep their connectivity better. In women, it is possible to observe a higher cavitation process, with loss of the number of trabeculae and their connectivity [48]. The result is a greater decrease in the trabecular surface in women than in men. Therefore, the greater periosteal apposition and the lesser loss of the trabecular surface makes the male bone thicker and more resistant to fractures.

The most important effects of periosteal bone apposition, greater in men, are the increase of cross-sectional area of the bone, so that load imposed per unit area decreases more in men than in women, and the lesser fall in volumetric density of the whole bone [48].

Interestingly, men with osteoporosis who undergo fragility fractures have greater loss of trabecular connectivity than men with osteoporosis without fractures [49].

In Table 2 we report the summarized results by Khosla et al. [50]. The authors detected the bone changes in the wrist by high-resolution 3-D pQCT imaging. From the table, it can be seen that the ratio between bone volume and total volume does not change differently between men and women, but the total number of trabeculae decreases significantly in women and not in men, while trabecular thinning is lower in women in the cancellous bone. In the evaluation of cortical parameters, we can see that the bone area does not change differently between the two sexes, the endosteal area increases significantly more in women, while cortical thinning is greater in men.

In men, testosterone levels are traditionally inversely correlated with fracture risk [51]. This demonstrates that the effect of androgens is anabolic on bone mass, with a stimulus on the periosteal apposition and therefore an increase in bone size and strength. However, androgens also have an important effect on the development of muscular mass and balance, reducing the risk of falling. The decrease in testosterone in men occurs more gradually than the decrease of estrogens in women, which decrease rapidly after menopause. The result is that the greatest growth in the osteoporotic fracture curve occurs in men about 5–7 years later than in women [52,53,54,55,56]. However, several cross-sectional and longitudinal evidence studies indicate that levels of bioavailable estradiol rather than testosterone are strongly correlated with the BMD and fracture risk in men [57,58]. Indeed, some of the biological actions traditionally attributed to testosterone acting via the androgen receptor (AR) may actually be dependent on its aromatization to estradiol (E2) through the activity of the CYP19A1 enzyme (aromatase). E2 ultimately binds the receptors α and βreceptor expressed on osteoblasts, osteoclasts, osteocytes, and marrow stromal cells [59]. Moreover, estrogen deficiency due to loss-of-function mutations in CYP19A1 (also known as aromatase deficiency) is associated with decreased bone mass, bone pain, increased bone turnover, and frequent fractures in men. The age-associated increase in sex hormone-binding globulin (SHBG) value may be the primary cause of the declining of sex steroid levels in men, that, in turn, produces a decline in bioavailable testosterone and estrogen levels (respectively, by 64% and 47% during the male lifespan) [57,58]. In fact, estrogen deficiency is one of the major reasons of osteoporosis in postmenopausal women. E2 deficiency markedly changes the balance of cells constituting the bone marrow, bone structure and bone resorption activity [60]. Given evolving understanding of the role of estrogen regulation of bone metabolism in men and the demonstration that their serum bioavailable (non-sex-hormone-binding, globulin-bound) estrogen levels decline with aging, estrogen decline also was believed to contribute substantially to the continuous bone loss of aging men. In both genders, estrogen decline increased bone resorption and also impaired compensatory increases in bone formation, although women manifest it at a younger age than men [61,62].

In a longitudinal study, Jones et al. calculated the bone loss per year with DXA at the femoral neck in a population of 769 men and women aged 60 years and older. The results showed that the decline in BMD (Bone Mineral Density) was 0.82% per year for men and 0.96% per year for women, with statistically significant difference. The rapid descent of the BMD curve occurred mostly between 65 and 69 years in women and between 74 and 79 years in men [63].

## 4. Causes of Osteoporosis in Men

In men, who, as already explained, the bone mass is greater than in women, it is necessary to address the diagnosis more often towards a secondary osteoporosis than in women. In women, 20 to 40% of osteoporosis is secondary to extraskeletal diseases, and this percentage raises until 65% in men [64,65]. Several risk factors and diseases have been shown to cause secondary osteoporosis in men. In the majority of cases, these factors increase the risk of osteoporosis even in women. The following is a list of causes that can lead to secondary osteoporosis in men [57,66,67]: alcoholism (defined as either daily intake, or greater than 10 servings per week), low BMI, glucocorticoid excess, hypogonadism (e.g., hormonal suppressive therapy for prostate cancer), hyperparathyroidism, hyperthyroidism, gastrointestinal disorders, such as malabsorption syndromes, inflammatory bowel diseases, gluten enteropathy, primary biliary cirrhosis, gastrectomy, hypercalciuria, chronic obstructive pulmonary disease, posttransplant syndrome, neuromuscular disorders, history of cerebrovascular accidents. Among the systemic illnesses, we find rheumatoid arthritis, multiple myeloma and other malignancies. Among the drugs inducing osteoporosis, we have glucocorticoids, anticonvulsants, thyroid hormone, chemotherapeutics. As regards lifestyle, we find cigarette smoking and sedentary lifestyle (Table 3). Black race (vs. white) was determined to be protective, as was a history of nephrolithiasis. Doubtful is the influence of physical activity, since it is assessed differently in the different studies [68,69,70,71,72]. It is indeed very difficult to compare the studies concerning this topic. There is no common criterion, for example, for evaluating high intensity and low intensity exercises, and it is hard to make a reliable comparison between two or more studies because the quantity and quality of the exercises are not standardized.

It has been estimated that approximately 20% of elderly men with osteoporosis have hypogonadism [73]. Hypogonadism, but also many of the other causes that induce osteoporosis in men are also involved in the development of sarcopenia [74]. The combination of osteoporosis and sarcopenia increases the risk of fractures, because of the reduced body balance caused by the loss of muscular strength, that facilitates falls, associated to the bone fragility due to osteoporosis.

### Monogenic Forms of Osteoporosis

It is now well known and widely demonstrated that osteoporosis has a hereditary component, and twin studies have shown genetic factors to determine up to 80% of its variance [75,76]. This inheritance is mostly attributable to multiple gene variants that combine with each other. Significant scientific advances have been made by studying rare monogenic forms of osteoporosis in which one mutation in a single gene with a major role in bone metabolism dominates and is alone sufficient to cause osteoporosis. The best known form is osteogenesis imperfecta, which can be caused by the alteration of one or more genes, but, recently, various other forms of monogenic osteoporosis have been found, such as autosomal dominant osteoporosis caused by WNT1 mutations or X-linked osteoporosis due to PLS3 mutations [77].

Interesting data on the genesis of gender-specific osteoporosis emerge from the studies on monogenic forms. In particular, we cite two papers that describe clinical cases of patients with congenital aromatase deficiency.

The aromatase enzyme complex catalyzes the conversion of androgens to estrogens in a wide variety of tissues, including the ovary, placenta, testis, brain, and adipose tissue [78].

It is known that estrogen deficiency is of crucial importance for the genesis of postmenopausal osteoporosis. The literature also shows that androgens are also an important protective factor for bone resorption in humans: men with hypogonadism have osteoporosis, and decreased serum testosterone concentrations in elderly men are a risk factor for fractures. Indeed, delayed puberty represents a factor risk for the decrease of BMD [79].

The clinical case reported by Smith et al. [80] is a 28-year-old male with a disruptive mutation of the estrogen-receptor gene. The patient had some physical characteristics and some pathologies related to the defect. Among these, the patient also suffered from osteoporosis. The subject suffered from severe osteoporosis with biochemical evidence of increased bone resorption. The estrogen resistance was concomitant to normal serum androgen concentrations. It is an important observation, because it demonstrates that androgens alone do not have the ability to induce skeletal maturation and avoid bone mass decrease, but estrogens have a fundamental role in bone mineralization, both in men and in women. What has probably happened in this man is that the high serum concentration of estrogens led to a compensatory increase in aromatase activity, which explains the normal serum concentration of androgens despite the secretion of luteinizing hormone. A study by Morishima et al. [81] confirms the hypotheses of Smith et al. on the decisive importance of estrogen on skeletal development and osteoporosis. The clinical case presented by the authors is represented by two siblings, a male and a female, who presented with an aromatase deficiency syndrome. Both siblings presented changes of the clinical feature and some pathologies consequent to hormonal alterations, which can be verified in the manuscript, but the data that may be significant in relation to the topic of this review are the following: in the XX sibling, the basal concentrations of plasma testosterone, androstenedione, and 17-hydroxyprogesterone were elevated, whereas plasma estradiol was low. In the XY sibling, the plasma concentrations of testosterone (2015 ng/dL), 5ordihydrotestosterone (125 ng/dL), and androstenedione (335 ng/dL) were elevated; estradiol and estrone levels were less than 7 pg/mL. Bone mineral densitometric indexes of the lumbar spine (cancellous bone) and distal radius (cortical bone) were consistent with osteoporosis; the distal radius was −4.7 SD below the mean value for age- and sex-matched normal men; indexes of bone turnover were increased.

From these observations it can be inferred that estrogens are essential for normal skeletal maturation and proportions (but not linear growth) in men as in women, the accretion and maintenance of bone mineral density and mass, and the control of the rate of bone turnover.

## 5. Diagnosis

The gold standard for the diagnosis of osteoporosis is unquestionably dual-energy X-ray absorptiometry (DXA), measured at the level of the femoral neck and possibly also at the vertebral level [82].

Although early studies concluded that men may fracture at higher BMD than women [83], larger, population-based studies have indicated that the opposite may be true. The Canadian Multicentre Osteoporosis Study proved that men followed for mean 8.3 years have lower rates of any major osteoporotic fracture (RR 0.75, 95% CI 0.60–0.93) and low trauma fractures (RR 0.75, 0.61–0.92) when controlling for age and lumbar spine BMD compared with women [84].

To date, the relationship between BMD and fracture risk is considered in men approximately the same as in women. However, there is no consensus for the screening of male osteoporosis or for the identification of people at high risk of fracture. Indeed, osteoporosis may be defined simply by the occurrence of a low-trauma fracture, by a BMD of 2.5 or more S.D. below the normal young mean as suggested by WHO, or by calculation of fracture risk.

The quantitative ultrasound (QUS) is mainly used for screening and its results must normally be confirmed with DXA, the more reliable method.

### 5.1. Current Recommendations for DXA Scan in Males

For males, it is recommended to use the same thresholds as females although the densitometric definition is not as well standardized as in postmenopausal women [85]. Therefore, the BMD is normally evaluated according to the WHO criteria, which remain valid for both men and women [83].

According to the 2012 Endocrine Society published guidelines for osteoporosis in men [86], all men should undergo DXA at 70 years of age. Under that age, measurement of BMD has to be done for all the men with risk factors. The risk factors considered by the Endocrine Society are listed in Table 4. The Endocrine Society GL recommends measuring BMD of the spine and hip. They suggest measuring forearm DXA when spine or hip BMD cannot be interpreted and for men with hyperparathyroidism or receiving androgen deprivation therapy for prostate cancer.

The ISCD (International Society for Clinical Densitometry) and NOF (National Osteoporosis Foundation) have the same recommendations [87,88]. Diversely, the United States Preventative Services Task Force did not recommend an established screening for men due to the consideration of insufficient evidence [89].

For patients younger than 50 years old, the ISCD recommends using a Z-score less than or equal to −2.5 D for the diagnosis of osteoporosis (which would compare males with peers of the same age if these data are available, or with women on the same age if male data are not available) [87]. Surprisingly, the prediction of fracture risk based on BMD seems more accurate in males than in females, especially in the elderly [90,91].

### 5.2. Radiology

In osteopenic osteoporotic patients who might have had undiagnosed vertebral fractures, thoracic and lumbar spine imaging should be obtained by Vertebral Fracture Assessment (VFA), since it has lower cost and radiation exposure than regular plain radiographs and can be obtained at the same time as the DXA scan. If VFA is not available, then lateral spine radiographs should be obtained [86]. Lateral radiographs of the spine should be done in all the patients admitted for a hip fracture. This procedure can create an organizational problem, which is not easily solvable in all the hospital.

### 5.3. Secondary Osteoporosis

The BMD, assessed through the bone densitometry, is in fact relative to the ‘quantity’ of bone. Bone strength, however, is the result of the ‘quantity’ added to the ‘quality’ of the bone. There are no reliable methods for evaluating the latter. Therefore, the densitometry should always be associated with a questionnaire which helps us in the identification of those men at risk despite normal BMD or T-score in the range of osteopenia. The most used is the FRAX (Fracture Risk Assessment Tool), released in 2008 by the World Health Organization. The FRAX provides for the collection of data relating to the patient, including all personal data (age, sex, race, weight, etc.) and the possible risk factors for fragility fracture.

In men, in particular, osteoporosis is often secondary [64,65]. Therefore, it is always necessary to investigate conditions that may have caused osteoporosis.

### 5.4. Laboratory Evaluation

If a specific cause of osteoporosis is suggested by history and physical exam, a more thorough investigation can be pursued, including free testosterone, prolactin, IGF-1, serum protein electrophoresis with free and light chains and/or urine protein electrophoresis, tissue transglutaminase antibodies, thyroid function tests, and PTH levels. In order to rule out endogenous hypercortisolism, 1 mg-dexamethasone suppression test is the preferred initial investigation, but two different essays are ideal to confirm endogenous hypercortisolism. The second test can be either free urinary cortisol (urine 24 h) or nocturnal salivary cortisol [88]. The laboratory tests suggested by the NOF are listed in Table 5.

Biochemical markers of bone turnover can also be checked at baseline to aid in risk assessment and to serve as an additional monitoring tool when treatment is initiated, also to assess compliance with treatment. However, their role in patient individual care is not well established.

### 5.5. Underscreening

As already pointed out above, male osteoporosis is underestimated, therefore also underinvestigated. In a study by Lim et al. [92], the authors evaluated the screening rate for 310 male patients aged 70 or over. Of these, who should all undergo DXA because of their age, only 11% had performed the exam and the majority of them were aged between 80 and 89 years. Antonelli et al. [93] carried out a study on patients aged 65 or older who had sustained a hip fracture: 12.1% of women and only 5.4% of men had undergone a DXA scan. Another study [38] evaluated 363 patients aged 50 years or older who had a history of atraumatic hip fracture. Only 27% of the women and 11% of the men had undergone a DXA scan within the 5 years prior to the fracture. These data show that, in general, osteoporosis and the risk of osteoporosis-related fracture is underestimated in both genders, but especially in men.

### 5.6. Recent Guidelines

In Table 6, the most recent guidelines on indications for screening of osteoporosis in men are listed. The analysis of these different guidelines shows that almost all the groups of study agree that older age is itself a risk factor for osteoporosis. The Endocrine Society [86], the NOF [17], the ISCD [87], the Brazilian Society of Rheumatology [46], all agree on the indication of DXA in all the individual over 70, while the Canadian Osteoporosis Society [94] recommends DXA at a younger age (subjects ≥ 65 years old), and the UK National Osteoporosis Guideline Group recommends first assessing fracture risks with existing tools in all individuals over the age of 50, giving indication for DXA even in individual with intermediate risk [9].

We agree with the guidelines proposed by the American Endocrine Society [86]: according to these recommendations, DXA should be screened in all men aged ≥ 70 and in all men aged 50 to 69 with risk factors (e.g., low body weight, prior fracture as an adult, smoking, etc.), evaluable also with the FRAX tool.

## 6. Treatment

The treatment of osteoporosis in men does not differ basically from that indicated for women. Actually, only few studies on the efficacy of anti-osteoporosis drugs in men have been carried out.

The most interesting studies about treatment, specifically addressed to osteoporosis in men, are summarized in Appendix A.

### 6.1. Non-Pharmacological Treatment

Non-pharmacological treatment of osteoporosis is essentially based on lifestyle behaviors and does not differ between men and women. Sufficient calcium intake and regular physical activity are strongly recommended from childhood. Avoiding alcohol and smoke is also very important.

Some observational studies found that physical activity as a separate factor can lead to an increase BMD [95,96], and improves muscular mass, strength, and resistance [97], although a sufficiently valid scientific demonstration of the cause-effect relationship is not easy. Moreover, some authors found that the physical activity-related BMD increase is approximately 2% in older men [95,96], but there is no strong evidence that physical activity might prevent fractures. There is no doubt that physical activity and targeted exercise programs reduce fall risk and allows to increase or maintain a good muscle tone and strength, thus reducing body instability and the risk of falls and fractures.

### 6.2. Pharmacological Treatment

#### 6.2.1. Supplement and Replacement Therapy

It has been widely demonstrated that calcium and vitamin D are essential for the treatment of osteoporosis and must always be associated with anti-osteoporotic drugs. Studies on women are numerous [98,99,100] while the literature on men offers us few specific studies. However, it is undoubted that supplementation with calcium and vitamin D in men should not be different from that recommended for women. The results of a large recent RCT conducted by LeBoff et al. [101], with a hybrid design with an overall cohort of 25,871 participants and a subcohort of 1054 participants, showed that supplemental vitamin D3 versus placebo had no effect on 2-year changes in aBMD and at the spine, femoral neck and total hip, or bone structure, regardless of gender. Some effects were seen in patients with baseline Vitamin D levels below the median (<14.2 pmol/L), in which there was a slight increase in spine aBMD (0.75% versus 0%; *p* = 0.043) and a reduced loss of total hip aBMD (−0.42% versus −0.98%; *p* = 0.044) with vitamin D3 supplementation.

Overall, a low calcium intake has been documented in women and men, independent of their bone health status. Data from subgroup analyses by sex showed that men’s intake is higher than women’s [102,103]. The Food Frequency Questionnaires (FFQs) may be considered quite convenient tools for the assessment of dietary calcium intake and their use should be promoted in clinical settings [104,105,106]. FFQs may inform people about their dietary habits and how these ones can be improved in order to increase calcium intake and the overall diet quality, and in turn to decrease the risk of osteoporosis and fractures. Nutritional education focused on healthy diets, including the consumption of foods rich in calcium and vitamin D, is highly desirable starting from childhood and mostly in post-menopausal women and older adults. Moreover, linked to the nutritional education, it would be useful to investigate their knowledge about sources of calcium and role of calcium on human health. This would allow to understand whether the low calcium intake is driven by a scarce awareness and would support the development of tailored nutrition education strategies.

A recent study by Min et al. [107] demonstrated that high-intensity physical activity with high serum vitamin D levels is associated with a low prevalence of osteopenia and osteoporosis.

#### 6.2.2. Bisphosphonates

Alendronate, risedronate, and zoledronic acid are approved for the treatment of primary osteoporosis in post-menopausal women, men, and glucocorticoid-related osteoporosis [108,109,110,111].

Actually, compared to the studies on osteoporosis in women, there are very few studies on pharmacological therapy in men, very few randomized controlled studies and most of the papers relate to populations with a low number of men. However, available evidence shows that the antiresorptive treatment (bisphosphonates and denosumab) increases bone density in osteoporotic men [110,112,113,114,115,116,117]. In two studies, the authors also demonstrated that alendronate reduces osteoporotic vertebral fractures [110,113], while one study showed the same effect on vertebral fractures by risedronate [115]. The study by Boonen et al. would disprove these findings [114]. A meta-analysis [118] showed instead that alendronate and risedronate significantly reduce the risk of vertebral fragility fractures, and that bisphosphonates also reduce the risk of non-vertebral fractures. Bisphosphonates are suggested for men, also in cases of secondary osteoporosis, for example in subjects with hypogonadism [119].

#### 6.2.3. Monoclonal Antibodies

Data on denosumab in male osteoporosis are scanty [120,121] but highlight how it also has the effect of increasing the femoral and vertebral BMD in the first year of treatment and of vertebral fractures also in the second year, in which instead the non-vertebral fractures lack of data.

Also, Denosumab was approved by use in men affected from osteoporosis. This occurred after the publication of the ADAMO study, which revealed an increase in BMD of the lumbar spine, femur, and distal radius after a treatment of osteoporosis in a population of 219 men [116].

The BRIDGE Trial (phase III) on the efficacy and safety of romosozumab in males showed the increase of BMD at both spinal and femoral level [122]. Romosozumab may therefore represent one of the drugs available for the treatment of osteoporosis in men in the next years. At present, although a phase III trial with 245 male subjects showed increased spine and hip BMD following 12 months of treatment with romosozumab, there is a safety concerns due to a low but significant increase in cardiovascular events in the medication group [122,123]. For this reason, FDA approval is only for postmenopausal women with no significant cardiovascular risk and it has not been approved for males yet.

In Table 7, evidence on anti-fracture efficacy of antiresorptive agents in men, is illustrated [112,113,115,124,125,126].

#### 6.2.4. Anabolic Drugs

A study about the use of teriparatide in men demonstrated its ability to increase BMD as early as 11 months [127]. The same drug was demonstrated to be able to reduce vertebral fractures in the 30 months following the discontinuation of treatment [126].

#### 6.2.5. Other Drugs

It is implied that, in the treatment and prevention of secondary osteoporosis, it is necessary to treat all the causes that may lead to or have lead to it. One of the most important causes of male osteoporosis is hypogonadism, both in young men (i.e., Klinefelter syndrome) and in aging. Therefore, to avoid excessive bone loss, testosterone replacement therapy becomes essential and is particularly recommended in young adults [119,128]. For patients with hypogonadism at higher risk for fracture or with contraindications for testosterone replacement, it is recommended to add nonhormonal pharmacologic therapy for osteoporosis. Indeed, it has been demonstrated that this category of hypogonadal patients are equally subjected to a high fracture risk, even after receiving adequate testosterone therapy for two years. Factors risks are the following: high-dose glucocorticoids therapy, frequent falls, history of a recent fragility fracture, particularly with a BMD T-score below −2.5 at any skeletal site, T-scores below −3.5 or even below −3.0 if they have other risk factors for fracture, T-score <−2.5 (or fragility fracture) [129].

### 6.3. Undertreatment

Feldstein et al. carried out a retrospective cohort study that included 1171 men aged 65 years and older [130], who had sustained a fracture. It is striking that only 16% of the patients with a hip or vertebral fracture received a medication for osteoporosis within the six months after the trauma, and treatment rates did not improve over time. Another study [131] evaluated the treatment for osteoporosis in male patients who had experienced distal radial fractures. The number of these patients who undertook osteoporosis therapy was negligible. Antonelli et al. [93] studied 417 patients discharged from the hospital for hip fractures. 23.3% of female patients and only 8% of male patients had received treatment for osteoporosis. Another study [38] shows that, after a hip fracture, in the period from 1 to 5 years after discharge, only 9% of men compared to 48% of women received adequate treatment for osteoporosis.

Therefore, most studies show that osteoporosis is undertreated in both genders, but much more in men, even in the secondary prevention of fragility fractures.

The Fr.Ost. (Osteoporotic Fracture, from Italian “FRattura OSTeopororotica”) is a prospective observational study we have been carried out within the Umbria region, Italy, from January 2015 to December 2020, by a team of geriatricians and orthopedists (RC, GR, CP), with the collaboration of 13 family doctors, whose project was approved by the Ethical Committee of the Umbria Region (CEAS). The aims of the study are: (1) to identify the prevalence of osteoporosis and fragility fractures in the general population; (2) to describe the clinical and functional characteristics of subjects with fracture risk; and (3) the type of treatment received over time. Participants were community-dwelling people aged 30 and more years, randomly selected 1 in 10 subjects within the lists of the participating family doctors. The preliminary and unpublished results of this study showed some interesting data, confirming the tendency towards underscreening, underdiagnosis, and undertreatment of osteoporosis especially in males. At baseline evaluation, the total number of subjects included in the study there were 1760. Of these, 963 were women, 797 men. The results showed that at least 366 men (45.9%) had risk factors for osteoporosis: 148 (18.5%) had at least one disease affecting bone metabolism, 191 (23.9%) were in treatment with one or more drugs affecting bone metabolism, and 27 (3.3%) had had at least one fragility fracture (9 hip, 8 rib, 5 wrist, 2 vertebral, 2 proximal humerus, 1 pelvis fractures) (Table 8). Despite the high number of patients who presented risk factors, only 9 (11%) had a diagnosis of osteoporosis made with DXA and only 13 (1.6%) were on anti-osteoporotic treatments. Of note, 6 patients were taking only vitamin D, 1 patient was being treated with a bisphosphonate, without calcium and vitamin D supplementation, and only 6 patients were receiving a combined treatment with bisphosphonate plus calcium and vitamin D (Table 9).

### 6.4. Adherence

Data about therapeutic adherence in male population are really limited.

One study found that one-third to two-third of men showed no adherence to bisphosphonates, resulting in an increased risk of fragility fractures [132].

The Denosumab Adherence Preference Satisfactions (DAPS) study [133], carried out in postmenopausal women, showed greater adherence to denosumab than bisphosphonate therapy, probably because patients had to undergo one subcutaneous injection every 6 months. It is assumed that the results obtained in women may be comparable to those in men. Bisphosphonates are suggested after stopping denosumab to reduce the increased risk of rebound-associated fractures [134].

## 7. Conclusions

The review of the literature regarding osteoporosis in men shows that few studies are dedicated to this issue. The paucity of literature on male osteoporosis is a sign that the latter is underestimated. Evidence shows that this condition is underdiagnosed and undertreated. This underestimation can have serious consequences, as epidemiological data tell us that, although it has a lower prevalence than osteoporosis in women, osteoporosis in men is not at all rare and presents high life risks. It is assumed that the diagnostic and treatment criteria are comparable to those usable for women, even if, in men, secondary osteoporosis has a higher frequency than the female counterpart.

The goal of all the physicians dealing with osteoporosis must be not to underestimate the risk that men may also be affected by this disease, and therefore to submit them to a screening as for women. We believe it is even more condemnable not to start the correct treatment for osteoporosis in male patients after a fragility fracture. This is, unfortunately, still common.

## Figures and Tables

**Table 1 ijms-22-02105-t001:** Summary of studies exploring mortality in men with osteoporosis.

Authors, Year	Study Design and Sample	Outcomes
Brown et al., 2021 [41]	Population-based retrospective 1:1 matched-cohort to controls using ICD-10 diagnostic codes for fractures from 1 January 2011 to 31 March 2015, in Ontario, Canada.	- Crude relative mortality risk 2.47- and 3.22-fold higher in matched fractured vs. non-fractured women and men, respectively. - 1 year absolute mortality risk post-fracture was 19.5% in men and 12.5% in women with fractures - absolute risk difference of 7.4% (95% CI 7.1–7.7%) in women and 13.5% (95% CI 12.9–14.0%) in men when compared to matched non-fracture controls
Lee et al., 2021 [42]	Korean National Health Insurance Research Database, we analyzed the cohort data of 24,756 patients aged > 60 years who sustained fractures between 2002 and 2013.	Mortality risk is higher in men then in women depending on the type of fracture: - **the first hip fracture** (HR, 2.25; 95% CI, 1.92–2.64 in women and HR, 1.96; 95% CI, 1.60–2.41 in men) - **the first vertebral fracture** (HR, 1.33; 95% CI, 1.15–1.53 in women and HR, 1.23; 95% CI, 1.01–1.48 in men) and the the number of subsequent fractures: in women - one, HR, 1.63; 95% CI, 1.48–1.80; - two, HR, 1.75; 95% CI, 1.47–2.08; - three or more, HR, 2.46; 95% CI, 1.92–3.15 in men - one HR, 1.42; 95% CI, 1.28–1.58; - two, HR, 2.03; 95% CI, 1.69–2.43; - three or more, HR, 1.92; 95% CI, 1.34–2.74
Bliuc et al., 2015 [34]	The Dubbo Osteoporosis Epidemiology Study prospective study Women and men ≥ 60 years followed from 1989 to 2011 with incident osteoporotic fractures (528 women and 187 men)	Similar distribution of fracture type in men and women - hip fracture (13% to 17%) - vertebral fracture (31% to 32%), - non-hip non-vertebral fracture (51% to 56%) RR of subsequent fracture is >2.0-fold for all levels of BMD - normal BMD: 2.0 (1.2 to 3.3) for women and 2.1 (1.2 to 3.8) for men; - osteopenia: 2.1 (1.7 to 2.6) for women and 2.5 (1.6 to 4.1) for men; - osteoporosis 3.2 (2.7 to 3.9) for women and 2.1 (1.4 to 3.1) for men. Post-fracture age-adjusted standardized mortality ratio is higher in men than women and increase with bone loss - osteopenia 1.3 (1.1 to 1.7) and 2.2 (1.7 to 2.9) for women and men, respectively, - osteoporosis 1.7 (1.5 to 2.0) and 2.7 (2.0 to 3.6) for women and men, respectively
Jiang et al., 2005 [35]	Population-based cohort of 3981 hip fracture patients ≥60 years admitted to hospitals in a large Canadian health region from 1994 to 2000	In-hospital mortality is 6.3%; 10.2% for men and 4.7% for women (adjusted odds ratio, 1.8; 95% CI, 1.3–2.4). Mortality at 1 year is 30.8%; 37.5% for men and 28.2% for women (adjusted *p* < 0.001)
Kiebzak et al., 2002 [38]	medical records from 363 patients (110 men and 253 women) aged 50 years and older with fragility hip fracture St Luke’s Episcopal Hospital between 1 January 1996, and 31 December 2000.	The 12-month mortality was 32% in men, compared with 17% in women (*p* = 0.003)
Center et al., 1999 [31]	5-year prospective cohort study in the semi-urban city of Dubbo, Australia, of all residents aged 60 years and older (2413 women and 1898 men).	Age-standardised mortality ratios are higher in men than in women for proximal femur (OR 3.17; CI95% 2.90–3.44 vs. OR 2.18; 95% CI 2.03–2.32); for vertebral sites (OR 2.38; 2.17–2.59 vs. OR 1.66; 95%CI1.51-1.80; and, for other major fractures (OR 2.22; 95%CI 1.91–2.52 and OR 1.92; 95% CI 1.70–2.14).
Diamond et al., 1997 [36]	Cohort study: 51 men aged ≥60 years and 51 age-matched women presenting to St George Hospital (a 650-bed tertiary care centre) with hip fractures, recruited retrospectively in 1995 from medical records and evaluated prospectively at 6 and 12 months after fracture.	14% men died in hospital compared with 6% of women (*p* = 0.06); men had more risk factors for osteoporosis (*p* < 0.01).

**Table 2 ijms-22-02105-t002:** Khosla et al., 2006 [50]. Summarized results which show the percentages of the trabecular and cortical parameters (see text). The population was made of 20-–90 years old subjects.

	Percentages	
	Women	Men
**Trabecular parameters**		
Bone volume/Total volume	−27%	−26%
Trabecular number	−13%	7%
Trabecular thinning	−18%	−24%
**Cortical parameters**		
Bone area	11%	11%
Endosteal area	27%	19%
Cortical thinning	−52%	−38%
Cortical vBMD	−22%	−16%

**Table 3 ijms-22-02105-t003:** List of risk factors for secondary osteoporosis in men [63,64,65,66,67]. The most part of these factors are common in men and and women.

Category	Risk Factor
**General**	Low BMI
**Lifestyle**	Alcoholism Smoke Sedentary lifestyle
**Hormonal**	Glucocorticoid excess Hypogonadism (e.g., hormonal suppressive therapy for prostate cancer) Hyperparathyroidism Hyperthyroidism
**Gastrointestinal disorders**	Malabsorption syndromes Inflammatory bowel diseases Gluten enteropathy Primary biliary cirrhosis Gastrectomy
**Systemic illnesses**	Rheumatoid arthritis Multiple myeloma Other malignancies
**Drugs**	Glucocorticoids Anticonvulsants Thyroid hormone Chemotherapeutics
**Other**	Hypercalciuria Chronic obstructive pulmonary disease Posttransplant syndrome Neuromuscular disorders History of cerebrovascular accidents

**Table 4 ijms-22-02105-t004:** Risk factors to assess in order to prevent fragility fractures in men, according to the Endocrine Society [86]. The FRAX assessment tool is also recommended.

Risk factors for Fragility Fractures in Men (*Endocrine Society*)
History of fracture after age 50
Delayed puberty
Hypogonadism
Hyperparathyroidism
Hyperthyroidism
Chronic obstructive pulmonary disease
Drugs such as glucocorticoids or GnRH agonists
Alcohol abuse
Smoking

**Table 5 ijms-22-02105-t005:** Laboratory tests suggested by the National Osteoporosis Foundation [17].

Laboratory Tests for Diagnosis of Secondary Osteoporosis in Men	
**Blood or serum**	▪ Complete blood count (CBC) ▪ Chemistry levels (calcium, renal function, phosphorus, and magnesium) ▪ Liver function tests ▪ Thyroid-stimulating hormone (TSH) +/− free T4 ▪ 25(OH)D ▪ Parathyroid hormone (PTH) ▪ Total testosterone and gonadotropin in younger men ▪ Bone turnover markers
*Consider in selected patients*	▪ Serum protein electrophoresis (SPEP), serum immunofixation, ▪ Serum-free light chains ▪ Tissue transglutaminase antibodies (IgA and IgG) ▪ Iron and ferritin levels ▪ Homocysteine ▪ Prolactin ▪ Tryptase
**Urine**	▪ 24-h urinary calcium
*Consider in selected patients*	▪ Protein electrophoresis (UPEP) ▪ Urinary free cortisol level ▪ Urinary histamine

**Table 6 ijms-22-02105-t006:** Most recent guidelines on indications for screening of osteoporosis in men.

Society	Year	DXA	X-rays
**Canadian Osteoporosis Society** [94]	2010	(1) Men ≥ 65 years (2) Men 50–64 years with risk factors (3) Men < 50 years with fragility fracture, prolonged use of glucocorticoids, use of other high-risk medications (i.e. aromatase inhibitors or androgen deprivation therapy), hypogonadism, malabsorption syndrome, primary hyperparathyroidism, other disorders strongly associated with rapid bone loss and/or fracture.	
**Endocrine Society** [86]	2012	(1) Men ≥ 70 (2) Men 50–69 yrs old with risk factors (history of fracture after age 50; diseases/conditions such as delayed puberty, hypogonadism, hyperparathyroidism, hyperthyroidism, or chronic obstructive pulmonary disease; drugs such as glucocorticoids or GnRH agonists; life choices such as alcohol abuse or smoking; or other causes of secondary osteoporosis)	
**NOF** [17]	2014	(1) Men age 70 and older, regardless of clinical risk factors. (2) Men age 50 to 69 with clinical risk factors for fracture. (3) Adults who have a fracture at or after age 50. (4) Adults with a condition (e.g., rheumatoid arthritis) or taking a medication (e.g., glucocorticoids in a daily dose ≥5 mg prednisone or equivalent for ≥3 months) associated with low bone mass or bone loss.	Vertebral X-rays: (1) All men aged 80 and older if BMD Tscore at the spine, total hip, or femoral neck is ≤−1.0 (2) Men aged 70 to 79 if BMD T-score at the spine, total hip, or femoral neck is ≤−1.5 (3) Men aged 50 and older with specific risk factors: ▪ Low-trauma fracture during adulthood (age 50 and older), ▪ Historical height loss of 1.5 in. or more (4 cm), ▪ Prospective height loss of 0.8 in. or more (2 cm), ▪ recent or ongoing long-term glucocorticoid treatment
**Brazilian Society of Rheumatology** [46]	2017	(1) Men ≥ 70 (2) Men < 70 with risk factors (glucocorticoids, alcoholism, hypogonadism, low body mass index (BMI), sedentary lifestyle, smoking, hyperthyroidism, hyperparathyroidism, malabsorption syndromes, chronic liver disease, hypercalciuria, anticonvulsants, immunosuppressants, organ transplantation, rheumatoid arthritis, multiple myeloma, mastocytosis	
**UK National Osteoporosis Guideline Group** [91]	2017	Assess fracture probability in men > 50 years who have risk factors for fracture. In individuals at intermediate risk, DXA should be performed.	
**International Society for Clinical Densitometry** [87]	2019	(1) Men > 70 years (2) Men < 70 years with risk factors for low BMD (low body weight, prior fracture, high risk medication use, disease or condition associated with bone loss).	

**Table 7 ijms-22-02105-t007:** Anti-fracture efficacy of drugs in men. All the papers mentioned are of evidence A.

	Anti-Fracture Efficacy in Men		
Antiresorptive Agents	Vertebral	Hip	Non Vertebral
Alendronate - 10 mg/daily - 70 mg/weekly	A [113]		
Risedronate - 5 mg/daily - 35 mg/weekly	A [124]	A [115]	A [124]
Zoledronate - 5 mg/yearly (EV)	A [112]		
Denosumab - 60 mg every 6 months (SQ)	A [125]		
**Anabolic agents**			
Teriparatide	A [126]		

**Table 8 ijms-22-02105-t008:** Characteristics of community living male adults enrolled in the FROST study.

Characteristics	Nr of Patients
Men (*n*,%)	797 (45.2)
Age (years, mean ± SD)	55.75 ± 15.5
Age classes (*n*,%)	
≤50	337 (42.2)
51–70	299 (37.5)
71–90	153 (19.1)
≥91	8 (1.0)
BMI (Kg/m^2^, mean ± SD)	26.25 ± 3.47
Parental osteoporosis and/or fractures (*n*,%)	49 (6.1)
Sedentary lifestyle (*n*,%)	347 (43.5)
Smoking habit (*n*,%)	
Current smokers	178 (22.3)
Former smokers	194 (24.3)
Alcohol intake (*n*,%)	
>4 unit/day	40 (5.0)
Dizziness (*n*,%)	38 (4.7)
Falls in the previous year (*n*,%)	17 (2.1)
At least one disease affecting bone metabolism (*n*,%) *	148 (18.5)
At least one drug affecting bone metabolism (*n*,%) ^#^	191 (23.9)
At least one fragility fracture (*n*, %)	27 (3.3)
Diagnosis of osteoporosis (*n*,%)	9 (1.1)
At least one antifracture treatment (*n*,%)	13 (1.6)

* The list of diseases affecting bone metabolism includes: diabetes, cancer, COPD, chronic kidney disease (KDIGO stage IV-V), inflammatory bowel diseases, gastrectomy, small bowel resections, malabsorption syndrome, rheumatoid arthritis or major rheumatic diseases, hyperthyroidism, multiple myeloma, celiac disease, primary hyperparathyroidism, hypogonadism. # The list of drugs affecting bone metabolism includes: steroids, aromatase inhibitors or androgen deprivation therapy, antiretroviral therapy, antiepilepctics, warfarin, PPI, thyroid hormones, loop diuretics, heparin, methotrexate, flutaininde, neuroleptics), hypogonadism, malabsorption syndrome.

**Table 9 ijms-22-02105-t009:** FROST study: on a total of 366 men at risk for osteoporotic fractures, only 11% received a diagnosis of osteoporosis and only 1.6% were on osteoporotic treatment. Of the latter, only 6 men out of 13 were on a combined therapy.

Population at Study		Diagnosis (DXA)	Anti-Osteoporotic Treatment
Total Number	1760 (963 Women, 797 Men)		
**Men with Risk Factors for Osteoporosis**	**Total**: 366 (45.9%) - One or more diseases affecting bone metabolism: 148 (18.5%) - In treatment with drugs affecting bone metabolism: 191 (23.9%) - One or more fragility fractures: 27 (3.3%)	9 (11%)	13 (1.6 %) - 6 only vitamin D - 1 only bisphosphonate - 6 combined therapy (bisphosphonate + vit. D + calcium)

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
