# Peer review of "Osteoporosis in Men: A Review of an Underestimated Bone Condition"

_ijms, 2021, doi:10.3390/ijms22042105_

Round 1

Reviewer 1 Report

This review provides a brief overview of male osteoporosis. Although I am not against this approach, several major limitations of this review are of major concerns: 

  1. How did the authors select the literature? Especially regarding the prevalence, economic burden of male osteoporosis. Were established guidelines by international bodies for diagnosis and treatment of male osteoporosis cited?
  2. When reporting the prevalence of male osteoporosis, the authors ought to report the country/city, age-range and the number of subjects involved, years of assessment. Just citing the prevalence numbers is not very informative to the readers. 
  3. What about the emerging secular trend of osteoporosis prevalence in developed/developing countries and the shift of the epic centre of osteoporosis in developing countries in Asia?
  4. I am surprised that age-related testosterone deficiency syndrome and androgen-deprivation therapy for prostate cancer treatment were not highlighted as a major cause of osteoporosis in men. 
  5. I think one of the outcomes of osteoporotic fracture that should be looked into is the reduction of quality of life. 
  6. Overall, I think much information is mentioned briefly without guiding the readers through what is new, what is lacking and what should be done on this important disease topic. 

Author Response

1. How did the authors select the literature? Especially regarding the prevalence, economic burden of male osteoporosis. guidelines by international bodies for diagnosis and treatment of male osteoporosis cited?
Thank you for your observation.
This is a narrative review presenting a brief overview of osteoporosis in men and discuss current knowledge about the evaluation and treatment of osteoporosis in males.
Since the literature on male osteoporosis is scanty, we read all the literature about that topic, especially relatively to the last ten years. We used the reference list of these articles to obtain additional sources. Among these articles, we selected those with the higher evidence and those with more interesting information, according to our opinion.
At your suggestion, we have added the guidelines of the most prestigious scientific societies.
2. When reporting the prevalence of male osteoporosis, the authors ought to report the country/city, age-range and the number of subjects involved, years of assessment. Just citing the prevalence numbers is not very informative to the readers.
Thank you for your suggestion.
We modified the manuscript by reporting the data you have suggested.
3. What about the emerging secular trend of osteoporosis prevalence in developed/developing countries and the shift of the epic center of osteoporosis in developing countries in Asia?
Thank you for your suggestion.
We added in the manuscript a part in which this topic is reported (lines 13-19).
4. I am surprised that age-related testosterone deficiency syndrome and androgen-deprivation therapy for prostate cancer treatment were not highlighted as a major cause of osteoporosis in men.
Thank you.
In line 151 and in lines 161-163, we mentioned those causes of osteoporosis. I do not know whether it is necessary to expand the part dedicated to this topic.
5. I think one of the outcomes of osteoporotic fracture that should be looked into is the reduction of quality of life.
I think this is important, but the reason why we did not dedicate a specific part of the manuscript to this topic, is the fact that the influence of osteoporotic fractures on the q.o.l. is not different in men and women. If you think it’s necessary, we’ll do it.
6. Overall, I think much information is mentioned briefly without guiding the readers through what is new, what is lacking and what should be done on this important disease topic.
Thank you for this consideration.
We hope that, with the numerous changes we have done to this manuscript, some more information is added.

Reviewer 2 Report

This is a well-written and timely review of osteoporosis in males.

Author Response

Thanks

Round 2

Reviewer 1 Report

Thank you for addressing some of my concerns. Specific comments for this round are as below: please do not capitalise the word "authors" in the text. line 45-50: The authors indicated that Thambiah and Yeap disprove the findings of Ballane et al., yet both papers show an increasing trend in fracture rate in developing countries. I suggest to rephrase the statement properly. I acknowledge that the authors have attempted to mention the populations of interest in epidemiological studies. However, there are still some leftover, for example: line 83: ref 13, line 94: ref 14, line 105-107: ref 25; line 109... an epidemiological observation: ref unknown; line 127: ref 14. line 115: There is only one study fall risk for men? line 141: ... warn not to trust DXA to evaluate the geometric characteristics of the bone... please kindly rephrase as the main function of DXA is not for geometric analysis. line 145-146: It sounds odd to use animals to indicate the relationship between body size and bone strength. There are plenty of studies which correlate bone density and body size in humans/men. line 148: replace 'prevalent' with prominent. line 180: replace 'prolonged period' with gradual line 183: I suggest the authors to mention about the conversation of testosterone to estrogen via aromatase enzyme first before mentioning estrogen deficiency impacts both sexes. By the way, how do you define estrogen deficiency in men. line 206: the influence of physical activities on bone health depends on the types of activities. Just mentioning it is different with studies is not very informative. section 4.1. I hope the case studies of mutation in ER or aromatase can be summarised. line 299, 301: replace 'mineralometry' with densitometry line 349: replace smoke with cigarette smoking. line 361: I don't agree with the statement " literature on men does not offer us specific studies worthy of note" Please read the following: https://asbmr.onlinelibrary.wiley.com/doi/full/10.1002/jbmr.3958 line 417: young adults with testosterone deficiency Table 8 doesn't show anti-fracture efficacy of drugs in men. Did the authors copy the table from another article? Please submit this article for proper proofreading and restructure more tightly.

Author Response

  • Please do not capitalise the word "authors" in the text.

Done, thank you.

  • Lines 45-50: The authors indicated that Thambiah and Yeap disprove the findings of Ballane et al., yet both papers show an increasing trend in fracture rate in developing countries. I suggest to rephrase the statement properly. I acknowledge that the authors have attempted to mention the populations of interest in epidemiological studies.

I’ve partially modified the text (lines 16-21)

  • However, there are still some leftover, for example: line 83: ref 13, line 94: ref 14, line 105-107: ref 25; line 127: ref 14.

Thank you. References were added.

  • line 109... an epidemiological observation: ref unknown;

The reference is nr 27.

  • line 115: There is only one study fall risk for men?

We are sorry, you’re right. We added more references on falls in men (lines 75-80).

  • line 141: ... warn not to trust DXA to evaluate the geometric characteristics of the bone... please kindly rephrase as the main function of DXA is not for geometric analysis.

Thank you for your suggestion.  Actually, the meaning of what we wrote was that DXA has not the ability to evaluate the geometric characteristics of the bone.  Anyway, I changed the sentence in order to make it more understable (I hope…): lines 103-104.  

  • Line 145-146: It sounds odd to use animals to indicate the relationship between body size and bone strength. There are plenty of studies which correlate bone density and body size in humans/men.

Thank you, I’m sorry, it was only a way to simplify the concept.  I removed that sentence.

  • line 148: replace 'prevalent' with prominent.

Thank you.  Correction done.

  • line 180: replace 'prolonged period' with gradual

Thank you.  I changed it in “The decrease in testosterone in men occurs more gradually than the decrease of estrogens in women” (line 138).

  • line 183: I suggest the authors to mention about the conversation of testosterone to estrogen via aromatase enzyme first before mentioning estrogen deficiency impacts both sexes. By the way, how do you define estrogen deficiency in men.

Thank you for your comment. We added a brief part about that topic (lines 140-159).

  • line 206: the influence of physical activities on bone health depends on the types of activities. Just mentioning it is different with studies is not very informative.

Thank you for your suggestion.  I tried to better specify the meaning of our statement (lines 182-186).  If you believe we should describe the cited papers more in detail, let me know.

  • section 4.1. I hope the case studies of mutation in ER or aromatase can be summarised.

Should we shorten the paragraph dedicated to this topic?

  • line 299, 301: replace 'mineralometry' with densitometry

Thank you.  Correction done.

  • line 349: replace smoke with cigarette smoking.

Thank you.  Correction done.

  • line 361: I don't agree with the statement " literature on men does not offer us specific studies worthy of note" Please read the following: https://asbmr.onlinelibrary.wiley.com/doi/full/10.1002/jbmr.3958

Yes, I modified the text by specifying that the published papers on men are few, and I cited the article you suggested (lines 349-358).

  • line 417: young adults with testosterone deficiency

I do not understand your comment.  Should we specify the normal value of testosterone?

  • Table 8 doesn't show anti-fracture efficacy of drugs in men. Did the authors copy the table from another article?

We did not copy the table, we just extracted the data from some articles and illustrated them schematically in the table, summarizing the data with the level of evidence (they are all RCTs).

Round 3

Reviewer 1 Report

Title: It contains two phrases and a full stop, which is quite uncommon. Suggest changing to "Male osteoporosis: a review of an underestimated bone condition."

Most of the comments are addressed, sans "using animals to indicate the relationship between body size and bone strength. There are plenty of studies which correlate bone density and body size in humans/men." The sentence is still seen at line 142.

The abstract contains capitalised "Authors". 

Line 112: "non-fatal injuries due to falls for women were 48.4% higher than the rates for men." please indicate the rate for men. 

I would still recommend the authors to seek professional proofreading service to improve the readability of the manuscript.

Author Response

  • Title: It contains two phrases and a full stop, which is quite uncommon. Suggest changing to "Male osteoporosis: a review of an underestimated bone condition."

Thank you. Correction done.

  • Most of the comments are addressed, sans "usinganimals to indicate the relationship between body size and bone strength. There are plenty of studies which correlate bone density and body size in humans/men." The sentence is still seen at line 142.

I’m sorry, I provided to remove that sentence.

  • The abstract contains capitalised "Authors". 

Sorry, I’ve corrected it.

  • Line 112: "non-fatal injuries due to falls for women were 48.4% higher than the rates for men." please indicate the rate for men. 

Thank you for your suggestion. I specified the rates.

  • I would still recommend the authors to seek professional proofreading service to improve the readability of the manuscript.

A proofreading service has already checked the manuscript.